# Performance-Based Design of Ferronickel Slag Alkali-Activated Concrete for High Thermal Load Applications

**DOI:** 10.3390/ma17194939

**Published:** 2024-10-09

**Authors:** Andres Arce, Anastasija Komkova, Catherine G. Papanicolaou, Thanasis C. Triantafillou

**Affiliations:** 1Department of Civil Engineering, University of Patras, GR-26504 Patras, Greece; andres@upatras.gr (A.A.); kpapanic@upatras.gr (C.G.P.); ttriant@upatras.gr (T.C.T.); 2Department of Civil, Environmental and Geomatic Engineering, ETH Zurich, 8093 Zürich, Switzerland

**Keywords:** alkali-activated material, geopolymer, concrete, design of experiment, olivine

## Abstract

This study aimed to develop optimized alkali-activated concrete using ferronickel slag for high-temperature applications, focusing on minimizing environmental impact while maintaining high compressive strength and slump. A response surface methodology, specifically the mixture design of experiments, was employed to optimize five components: water, FNS-based alkali-activated binder, and three aggregate sizes. Twenty concrete mixes were tested for slump and compressive strength before and after exposure to 600 °C for two hours. The optimal mix achieved 88 MPa compressive strength before heat exposure and 34 MPa after, with a slump of 140 mm. An upscaled version with improved workability (210 mm slump) maintained similar unheated strength but showed reduced post-heating strength (23.5 MPa). Replacing limestone with olivine aggregates in the upscaled mix resulted in 65 MPa unheated and 32 MPa post-heating strengths. Life Cycle Analysis revealed that the optimized ferronickel slag alkali-activated concrete’s CO_2_ emissions were 77% lower than those of ordinary Portland cement concrete of equivalent strength. This approach demonstrated the applicability of mixture design of experiments as an alternative design methodology for alkali activated concrete, providing a valuable performance-based design tool to advance the application of alkali-activated concrete in the construction industry, where no prescriptive standards for alkali-activated ferronickel concrete mix design exist. The study concluded that the developed ferronickel slag alkali-activated concrete, obtained through a performance-based mixture design methodology, offers a promising, environmentally friendly alternative for high-strength, high-temperature applications in construction.

## 1. Introduction

As a result of the extensive use of concrete, the construction sector contributes 1.28 billion tons of CO_2_ annually [1]. The production of cement alone accounts for 5% to 8% of total global emissions [2]. Concrete stands as the second most consumed material worldwide; its use surpasses the combined consumption of all other construction materials. Given the current global challenges associated with climate change, it is imperative to reduce our carbon footprint. The construction sector holds a pivotal role in this effort. One promising avenue for the concrete industry is the adoption of alternative concrete production technologies that offer a reduced environmental impact. Alkali activation stands out among these technologies; concrete produced through alkali activation has been shown to reduce CO_2_ emissions by up to 80% [3] compared to ordinary Portland cement (OPC).

Alkali-activated materials (AAM) are synthesized by activating an aluminosilicate precursor with an alkaline solution. Renowned for their outstanding durability, these materials exhibit stability at elevated temperatures and resist chemical attacks [4]. The technology has been found to have extensive applications in the production of alkali-activated concrete (AAC). Various methodologies exist for AAC design, ranging from the adaptation of established prescriptive methods initially developed for Portland cement concrete (PCC) [5] to the utilization of cutting-edge approaches like machine learning for optimizing mix proportions [6].

Concrete mix design entails determining the optimal ratio between coarse granular materials and a matrix that both fills the voids between these particles and binds them together. In this study, the authors chose to devise an optimal concrete mix design utilizing a statistical tool known as design of experiments (DOE). Specifically, the mixture design of experiment (MDOE) is a response surface methodology (RSM) technique that facilitates the formulation of an optimal ingredient balance based solely on their respective proportions. Although MDOE initially gained traction in the chemical, pharmaceutical, and food industries, its application in the construction sector has emerged more recently [7]. A notable advantage of mixture design is its efficiency. Once a test matrix is defined and results are obtained, models can be developed to correlate performance indicators with ingredient proportions. Although the formulation of these models often necessitates the creation of numerous samples, the ability to establish a relationship between concrete characteristics and their constituents is invaluable. This relationship enables the generation of multiple mix designs tailored to optimize specific parameters.

Previous studies have utilized DOE techniques to address the challenges associated with alkali-activated concrete mix design. Simsek et al. [8] optimized and modeled the mixture proportions of C50 concrete using a full factorial design (another type of DOE). Their study highlighted the necessity of a systematic methodology to optimize and model quality characteristics in ready-mix concrete, given its many conflicting factors. The authors demonstrated that DOE effectively optimized the mix proportions and met the expected properties of C50, thereby providing a valuable tool for improving system performance in concrete manufacturing.

Kanta et al. [9] also employed DOE to optimize alkali-activated slag concrete, particularly focusing on early-age compressive strength. Their statistical analysis validated the method’s effectiveness, with optimized mixes yielding compressive strengths of approximately 53 MPa at 28 days. The study highlighted the importance of independent variables in influencing compressive strength and demonstrated the potential of DOE for accurate predictions and optimizations. Mendes et al. [10] demonstrated the utility of mixture design experiments in optimizing alkali-activated materials composed of industrial waste, such as chamotte and waste glass. Their findings showed that while waste glass increased porosity and reduced compressive strength, its interaction with chamotte resulted in a synergistic effect beneficial for alkali-activation. Finally, Moseson et al. [11] optimized a ground granulated blast furnace slag (GGBFS) cement mix using DOE, achieving significant reductions in CO_2_ emissions and energy consumption. Their study showed that an optimized mix could produce only 25 kg CO_2_/ton, a 97% reduction compared to ordinary Portland cement (OPC), and could be both cost- and performance-competitive.

Despite the existing body of literature on the application of DOE in concrete mix design, it is noteworthy that, at the time of the authors’ research, no prior studies had specifically endeavored to develop an alkali-activated concrete mix tailored for high-temperature applications using the design of experiments.

Recent research has demonstrated the significant potential of ferronickel slag (FNS) as a raw material for the production of AAM and geopolymers, thus contributing to more sustainable construction practices. Ferronickel slag is a by-product of the pyrometallurgical treatment of laterites in the production of ferronickel, particularly from the reductive smelting processes in electric arc furnaces. Studies have shown that FNS can effectively be utilized in the synthesis of inorganic polymers, providing mechanical properties comparable to traditional cementitious materials. For instance, Maragkos et al. [12] investigated the synthesis of geopolymers using FNS, optimizing synthesis parameters to enhance mechanical properties. Their findings indicated that geopolymers produced under optimum conditions achieved compressive strengths of up to 118 MPa with extremely low water absorption (0.7–0.8%). The research identified that the compressive strength of the produced materials was influenced by the concentration of sodium hydroxide (NaOH) in the activation solution, with an optimal concentration of approximately 7 M being necessary to balance alkaline dissolution and polycondensation phenomena. Similarly, Komnitsas et al. [13] examined the alkali activation of Polish ferronickel slag and reported that under specific conditions (e.g., NaOH molarity of 8 M, curing temperature of 80 °C), the produced inorganic polymers exhibited compressive strengths exceeding 65 MPa. These AAMs also demonstrated excellent structural integrity when immersed in water and acidic solutions, highlighting their durability.

Recent research has shown promising results for the use of FNS as an alternative precursor in the development of alkali-activated materials. Kim et al. [14] explored the substitution of FNS for OPC in concrete, focusing on the effects of FNS fineness and replacement ratios. The research highlighted that, although FNS concrete exhibited delayed strength development at early ages, long-term strength was comparable to OPC concrete due to the latent hydration properties of FNS. Moreover, the study emphasized the enhanced durability of FNS concrete, particularly in resistance to chloride penetration and sulfate attack. This improvement was attributed to the formation of a denser pore structure, which restricted the ingress of harmful ions. However, the increased carbonation risk at higher FNS replacement ratios was noted as a potential drawback. Kim et al. also highlighted the environmental benefits of FNS concrete, including reduced CO_2_ emissions during production and lower raw material costs compared to conventional supplementary cementitious materials like ground granulated blast furnace slag (GGBFS). Additional studies have also examined the replacement of GGBFS in concrete with ground ferronickel slag (GFNS) [15,16], as well as its use as a replacement in OPC [17]. There has also been notable research interest in the application of FNS as a fine aggregate [18,19].

In this study, the authors utilized ferronickel slag—a readily available ferrosialate precursor in northern Greece—to explore the formulation of an alkali-activated concrete mixture characterized by minimal environmental impact and enhanced durability, particularly under high-temperature conditions. Beyond employing GFNS as a primary component of the concrete matrix, the authors incorporated coarse ferronickel slag (ranging from 0 mm to 4 mm) as a fine aggregate. The substantial utilization of ferronickel slag in this research aligns with the authors’ commitment to advancing circular economy principles within the construction sector. In Greece, while the scarcity of sand is becoming a concern, approximately two million tons of FNS are generated annually, with only 30% being recycled; the remainder is relegated to landfills or submerged in the sea [20]. The mismanagement of FNS not only results in wastage but also incurs an annual disposal cost of EUR 650,000 (based on 2007 metrics).

Ferronickel slag has emerged as a promising material for developing alkali-activated materials suitable for high-temperature applications, primarily due to its unique properties and advantages over traditional binders. Recent studies, such as those by Sakkas et al. [21], demonstrated that FNS-AAM can achieve remarkable mechanical and thermal properties, including high compressive strength and low water absorption. Specifically, the research indicates that FNS geopolymers can reach compressive strengths up to 120 MPa, significantly outperforming conventional concrete materials. Additionally, these geopolymers have shown excellent fire resistance characteristics, making them suitable for applications requiring passive fire protection. For instance, the materials were tested against standardized fire resistance tests, demonstrating structural integrity and thermal stability under high temperatures.

Furthermore, research conducted by Zhang et al. [22] confirmed the feasibility of using FNS as a refractory material, with geopolymers exhibiting impressive thermal resistance, reaching refractoriness of up to 1270 °C. This study showed that the mechanical strength of FNS-based materials is maintained even after exposure to extreme thermal conditions, with a residual compressive strength of 59.65 MPa after treatment at 1100 °C. Such findings suggest that FNS not only meets the high-performance criteria required for construction applications but also enhances the longevity and durability of structures exposed to extreme heat.

In the work by Arce et al. [23], the use of FNS in conjunction with a design of experiments approach allowed for the optimization of AAM formulations tailored for high thermal load applications. The study highlights the importance of minimizing chemical activators, which aligns with environmental sustainability goals by reducing CO_2_ emissions by approximately 55% compared to traditional cement. The optimal mix design achieved high unheated compressive strengths and favorable residual strengths after thermal exposure, showcasing FNS’s potential in developing sustainable construction materials that do not compromise performance.

The present study not only presents a methodology for AAC mix design but also addresses the knowledge gap concerning the use of ferronickel slag as the main binder in AAC, which had not been explored prior to this work. Additionally, by incorporating GFNS as a fine aggregate, this research enhances slag reutilization, thereby minimizing the environmental impact of slag landfilling and supporting circular economy principles.

## 2. Materials and Methods

### 2.1. Raw Materials

Ground ferronickel slag served as the primary ferrosiliate source material, while sieved ferronickel slag was utilized as fine aggregate (ranging from 0 mm to 4 mm). The FNS was generously provided by The General Mining and Metallurgical Company SA in Larissa, Greece. As a supplementary silica source, silica fume with a silicon oxide content exceeding 85% was incorporated. The particle size distributions of both GFNS and silica fume (SF) were meticulously characterized using laser diffraction analysis via a Malvern Mastersizer 2000 produced by Malvern Panalytical in Worcestershire, UK. The d50 and d90 values for GFNS were determined as 8.36 μm and 29.1 μm, respectively, while those for SF were 12.87 μm and 29.98 μm, respectively. Coarse limestone aggregates were employed in two distinct sizes: 4–8 mm and 8–16 mm. The alkaline solution was meticulously prepared by blending water with potassium silicate (KS) and potassium hydroxide (KOH). KS, acquired as a proprietary solution named Geosil^®^ 14517 (produced by Wöllner, based in Ludwigshafen, Germany), boasted a dry content of 45% and exhibited a modulus of 1.6. Meanwhile, KOH was sourced in pellet form with a purity level of 90%.

The chemical compositions of both GFNS and SF were analyzed using X-ray fluorescence (XRF) spectroscopy, results are provided in Table 1. This analysis encompassed the measurement of both major elements, including SiO_2_, Al_2_O_3_, CaO, MgO, MnO, Fe_2_O_3_, K_2_O, Na_2_O, P_2_O_5_, and TiO_2_, as well as minor elements. For the analysis, 1.8 g of the dried ground sample was combined with 0.2 g of wax, which served as a binder. This mixture was subsequently mounted onto a 32 mm diameter circular powder pellet by pressing onto a base of boric acid. The XRF analysis was conducted using a RIGAKU ZSX PRIMUS II spectrometer (provided by Applied Rigaku Technologies, Inc., Austin, TX, USA), equipped with an Rh-anode operating at 4 kW, ensuring precise analysis of both major and trace elements. The spectrometer was outfitted with a range of diffracting crystals, including LIF (200), LIF (220), PET, Ge, RX-25, RX-61, RX-40, and RX-75, to facilitate comprehensive elemental analysis.

Binder was defined as the sum of GFNS, SF, KOH, and the dry part of KS. The author previously studied the optimal ratio of these ingredients [23] and found it to be 846.1 kg GFNS, 62.4 kg SF, 27.2 kg KOH, and 64.4 kg KS for 1 ton of binder. These proportions were kept constant through all 20 mixes used for the calculation of the optimal mix. The binder composition includes the dry parts of KS and KOH, which were added as part of the chemical activator solution. The percentage limits for the binder, as outlined in Table 2, can be converted into the corresponding concentration ranges for KOH and KS in the final mixes. These concentrations range from a minimum of 1.8 mol/L of KOH and 1.22 mol/L of KS to an upper boundary of 2.06 mol/L for KOH and 1.4 mol/L for KS. The selection of these boundaries was based on the results of the author’s previous binder optimization study presented in [23], supplemented by empirical observations from preliminary mix trials conducted before the main experimental campaign.

The binder was then combined with FNS sand (0–4 mm), faucet water, and two limestone aggregates (4–8 mm and 8–16 mm). These five component proportions varied within the limits reported in Table 2. Additionally, this table reports the on-carbon emissions and energy consumption of each component. CO_2-eq._ emissions were calculated using SimaPro v8.5 software, Ecoinvent v3.4 database.

Outside the scope of optimum mix design, olivine aggregates were also tested to study their potentially better thermal compatibility with the GFNS matrix. The olivine aggregates used in this work were quarried in Northern Greece.

### 2.2. Concrete Production and Specimens Preparation

The aluminosilicate materials detailed in Section 2.1 were activated using an alkaline solution formulated from KS and KOH. This solution was prepared approximately 24 h prior to the casting of the concrete. To create the alkaline solution, water was first added to a deep container—this depth was crucial to prevent potential spills of the corrosive solution. Subsequently, Geosil^®^ 14517 (a combination of KS and water) was introduced into the container, followed by the addition of KOH pellets. The mixture was then vigorously stirred using a long stirrer. Given that the dissolution of KOH pellets is exothermic, the solution was allowed to cool naturally for a period of 24 h.

The limestone-based ferronickel slag alkali-activated concrete (L-FNS-AAC) specimens were produced in 60 kg batches using a 70 L electric pan-type concrete mixer. The production process started with adding dry components from the highest to the lowest size into the mixer. It was observed that this order facilitated the dry mixing process. Once all dry ingredients were placed, the electric mixer was used for 30 s, ensuring a homogeneous mixture. At this point, the activator solution (water + KOH + KS) was gradually added, and hand mixing followed (for approximately 20 s) to avoid splashing of the solution and promote absorption by the solid components. The final mixing phase involved operating the electric mixer for 10 min, with a pause after 5 min to scrape off any residual unmixed ingredients adhering to the mixer walls.

Each concrete batch was cast into six steel molds, each measuring 150 mm on each side. Following casting, the molds were subjected to vibration using a vibration table for a duration of 30 s to ensure optimal compaction. To minimize evaporation, the concrete cubes were then shielded with a thin plastic sheet. The demolding process was conducted the subsequent day.

### 2.3. Fresh- and Hardened-State Properties Tests

The slump test, in accordance with EN 12350-2 [24], was conducted immediately upon the completion of the mixing process to assess the workability of the concrete. For the curing phase, the specimens were allowed to mature under laboratory conditions for a period of 28 days. These conditions were maintained at a temperature of 20 ± 2 °C and a relative humidity of 65%. Throughout the curing period, the specimens were wrapped in multiple layers of plastic foil to prevent moisture loss. No additional heat curing was employed.

For compressive strength evaluation, the concrete cubes were subjected to compression tests using a compression testing machine equipped with a maximum load capacity of 4000 kN. The application of load followed the guidelines specified in ASTM C 39 [25], with a constant rate of 1 mm/min.

The residual compressive strength after unstressed heating conditions and cooling is typically considered to be lower than the compressive strength obtained at a “hot” state after stressed or unstressed heating conditions [26,27]. Hence, the residual compressive strengths reported in this work comprise a conservative lower bound for the material compressive strength during heating.

### 2.4. Design of Experiment (DOE)

The optimal proportions of the five components (A: binder, B: FNS sand, C: coarse aggregates 4–8 mm, D: coarse aggregates 8–16 mm, and E: faucet water) were investigated through the mixture design of experiment. Mixture DOE is used to find the optimal proportions of a mixture based on a multi-response desirability function. For the present study, the selected responses were slump and compressive strength of the cubes, both unheated and heated.

To facilitate calculations, the authors utilized Design Expert^®^ v11.1.2.0. The optimization of components through mixture DOE starts with the selection of component constraints. The following constraints were considered: the Water-to-binder ratio was set to range between 0.235 and 0.27; the sand-to-binder ratio between 1 and 3; the solid-to-liquid ratio from 13 to 17; aggregates were 0–4 mm and 8–16 mm, which were both restricted to a range from 10% to 50% of mortar (the sum of water, binder, and FNS sand) content, by weight. These restrictions were selected based on preliminary experimental work (not presented here) and intended to produce concretes with neither excessive nor non-existent flow. The performance evaluation of each concrete mixture was conducted through slump tests and compressive strength tests, both pre- and post-heat exposure.

The D-optimality criterion was employed to formulate 20 distinct mix designs using Design Expert^®^, while adhering to the aforementioned constraints. Six concrete cubes were fabricated from a singular batch for each mix design. This setup facilitated the acquisition of three compressive strength measurements under ambient (unheated) conditions and an additional three under elevated (heated) conditions. Post-heating, the cubes were tested subsequent to cooling to ambient temperature. A singular slump value was determined immediately after the conclusion of the mixing process. A comprehensive summary of all mix designs, including their respective compositions per cubic meter, is delineated in Table 3.

### 2.5. Heating Regime

For each mix design, three concrete cubes underwent thermal exposure in a controlled environment. These cubes were subjected to a temperature of 600 °C within a 0.125 m^3^ electric furnace. Limestone aggregates undergo a chemical transformation into calcium oxide (CaO) and carbon dioxide (CO_2_), which begins at temperatures as low as 600–700 °C. Exceeding this temperature results in severe damage to the aggregates, an issue the authors observed during experimental trials prior to the main study. The exposure temperature of 600 °C was selected to prevent this effect from occurring.

The heating protocol involved a temperature increase of 5 °C per minute until the target temperature was achieved. Upon reaching the target temperature, the heat was maintained for a duration of 2 h. The selection of the heating rate and duration of thermal load exposure follows an established trend in studies [28,29,30] on the thermal behavior of alkali-activated materials, where heating rates typically range from 1 to 10 °C/min and the exposure duration from 1 to 4 h.

Subsequent to the 2 h heating phase, the electric furnace was deactivated, allowing the specimens to undergo natural cooling over a span of approximately 24 h, during which the furnace’s shutter remained closed.

### 2.6. Life Cycle Assessment (LCA) Method

Adhering to the ISO 14040 LCA framework [31], this analysis aimed to evaluate the environmental footprint associated with the proposed alkali-activated concrete mixes and offer a comparison to OPC-based concrete variants that exhibit comparable mechanical properties. The functional unit chosen for this assessment is 1 m^3^ of concrete.

The system boundaries for the LCA encompass the environmental impacts stemming from the production and pre-treatment stages of the constituent materials. To conduct this evaluation, the SimaPro 8.5 software was utilized. Life cycle inventories (LCIs) pertinent to the production processes of raw materials were sourced from the Ecoinvent v3.4 database.

The GFNS concrete developed in this study was evaluated under two distinct scenarios based on the ferronickel slag classification: a waste material and an industrial by-product. In the former scenario, emissions attributable to the slag’s production via the furnace process were not allocated to its manufacturing phase; only the pre-treatment stage was factored into the analysis. Conversely, in the latter scenario, an economic allocation approach was adopted [23].

To quantify the global warming potential (GWP) of the proposed concrete mixes, the IPCC2013 life cycle impact assessment method was employed.

## 3. Results and Discussion

Figure 1 presents the test results for both fresh and hardened states for the mixes described in Table 3.

### 3.1. DOE Models

The test results presented in Figure 1 were input into the Design Expert^®^ software. Nine out of 140 data points were identified through various diagnostic plots generated by Design Expert^®^ software. These plots included normal plots, Cook’s distance, and Residual vs. Predicted values plots. After data curation, the refined dataset was used to develop regression models correlating slump and compressive strength (heated and unheated) with the component’s proportions. These regression equations are listed in Table 4. To use these equations for prediction, users can substitute the corresponding component contents (designated as A, B, C, D, and E in Table 2) in grams. The sum of all components corresponds to 60,000 g (or 60 kg). The coefficients for each model (Table 4) were selected by testing adjusted R^2^ criteria, Akaike, and Bayesian information criteria (AIC and BIC), and selecting the criteria that resulted in the highest predicted R^2^. This procedure was executed one by one for each model corresponding to one of the three studied responses.

Table 5 presents the fit statistics for each model, including mean values, standard deviation (Std. Dev.), and coefficient of variation (CoV). Three types of R^2^ values are provided: the conventional R^2^, as well as the adjusted and predicted R^2^ values. The discrepancy between the predicted and adjusted R^2^ values was found to be less than 20%, which is considered a reasonable level of agreement. Additionally, the adequate precision metric, which evaluates the signal-to-noise ratio, exceeded a value of 4 in all models. A value of 4 serves as a lower threshold, indicating an 80% probability that the predicted response is attributable to the influence of the ingredients rather than natural variation, also known as noise [7].

### 3.2. Optimal Concrete Mix Design for High Temperature Applications

Upon deriving a regression equation for each of the three responses, all the required input to calculate an optimal mix was obtained. The objective was to identify the “sweet spot” where the component proportions would yield both slump values closely aligned with the S2–S3 classes and high compressive strengths. To achieve this, a desirability function, facilitated by the Design Expert^®^ software, was employed. This function enabled the calculation of a desirability coefficient for each one of the multiple responses. Subsequently, these coefficients were combined to produce an overall desirability coefficient, reflecting the mix’s overall performance. An importance factor ranging from 1 to 5 was assigned to each response to account for its relative significance in determining the optimal mix. In this study, slump and unheated compressive strength were assigned a factor of 4, while heated compressive strength was assigned a factor of 5.

Utilizing the desirability algorithm in conjunction with a hill-climbing algorithm, Design Expert^®^ was used to identify the concrete mix design that yielded the highest desirability coefficient. The ingredient proportions for 1 m^3^ of this mix are detailed in Table 6. A comparison between the predicted and measured values of slump and compressive strength in the table reveals a high level of agreement, with errors below approximately 3%. This level of error is notably lower than the discrepancies reported in other studies. For instance, Moseson et al. [11] reported a difference of around 15% between expected and validated results when optimizing AAC mixtures for CO_2_ emissions reduction using GGBFS activated with sodium carbonate through mixture design of experiments (DOE). Similarly, Kanta et al. [9] observed a model prediction error of approximately 9% when predicting the mechanical properties of AAC based on steel slag and foundry sand, employing response surface methodology (RSM) and Minitab^®^ software.

The GFNS AAC mix was further refined with the objective of adapting it for upscaled concrete castings, where a more fluid consistency would be advantageous. Consequently, an additional mix was formulated with the goal of increasing the slump to 200 mm. This modified mix design is detailed in Table 6. As evident from the table, alterations in mix proportions were minimal, with the greatest change amounting to 6.3%. Special attention was devoted to maintaining the chemical activator content nearly constant to prevent a significant increase in CO_2_ emissions associated with the modified mix. The resultant mix exhibited a slump of 210 mm and an unheated compressive strength nearly identical to that of the optimal DOE mix but with a 30% lower heated strength, likely attributable to the increased water content. The mix demonstrated sufficient workability for up to 3 h following mixing.

The results of the optimal mix design demonstrate a significant breakthrough in the formulation of alkali-activated concretes. By employing a mixture design of experiments, a mix was developed with notably low potassium hydroxide (KOH) concentrations at 1.91 mol/L and potassium silicate (KS) at 1.31 mol/L. This study’s molar concentration of chemical activators is substantially lower than that of similar research. For example, Kong and Sanjayan [32] developed several concrete mixes that achieved approximately 65 MPa in compressive strength by activating fly ash with a KOH solution at 7.0 M, relying on heat curing at 80 °C for 24 h. Aslani and Asif [33] studied the production of geopolymer concrete for high-temperature applications using ground granulated blast furnace slag and fly ash, producing concrete mixes of approximately 40 MPa without heat curing by using an activator solution with a NaOH concentration of 14 mol/L (seven times higher). Similarly, Mehta et al. [34] reported the production of alkali-activated concrete based on fly ash and ground granulated blast furnace slag, achieving approximately 40 MPa using an activator solution of 10 mol/L of sodium hydroxide.

The mixture design of the experimental approach used in this study capitalized on non-linear component interactions, enabling the optimization of component proportions that are challenging to achieve with the one-factor-at-a-time modification approach. Through this methodology, the study produced an upscaled optimal concrete mix with a high compressive strength of 85 MPa and a slump of 210 mm while maintaining minimal chemical activator use and eliminating the need for heat curing or additives. In addition to its high mechanical performance, the optimal upscaled mix maintained workability for up to 3 h after mixing. The low concentrations of chemical activators help to minimize operational challenges by reducing risks such as corrosiveness and high heat release associated with the exothermic reactions during alkaline activator solution production. The optimal design, focused on minimizing chemical activators, also contributed to lower CO_2_ emissions and costs, as these activators are significant contributors to both environmental impact and expenses in alkali-activated materials. These findings highlight the effectiveness of the mixture design of experiments methodology in producing sustainable and high-performance alkali-activated concrete suitable for the construction industry.

### 3.3. Olivine Aggregate Replacement in the Optimal Concrete Mix Design

In an attempt to enhance the residual compressive strength following heating, the limestone coarse aggregates reported in the upscaled mix design detailed in Table 6 were replaced with olivine aggregates. The resulting olivine aggregate FNS-based AAC (O-FNS-AAC) exhibited a slump of 200 mm, which closely mirrors that of its limestone aggregate counterpart. Additionally, it demonstrated unheated and heated compressive strengths of approximately 65 MPa and 32 MPa, respectively, when heated to 600 °C for 2 h. The reduction in unheated compressive strength compared to the limestone aggregate AAC is attributed to the flakiness of the olivine aggregates and their suboptimal grading. Notably, the relative residual compressive strength of O-FNS-AAC (expressed as a percentage of the unheated strength) surpassed that of limestone aggregate AAC by a factor of approximately two. Furthermore, the absolute value of the residual compressive strength was greater for O-FNS-AAC than for the upscaled limestone-FNS-AAC mix. This higher absolute value suggests an improved mitigation of thermal strain mismatch between the matrix and aggregates.

### 3.4. Fresh State Properties: Slump

Many of the DOE mixes yielded zero slump, a condition likely attributable to the stringent limits set for water content (ranging from 5.6% to 6.3% by weight). Additionally, this outcome can be associated with the elevated silica modulus (1.26), which Law et al. [35] identified as a contributing factor to slump loss. Figure 2 illustrates an example of a zero-slump mix (mix 8), side to side with a mix exhibiting a 70 mm slump (mix 3).

Figure 3 presents contour plots corresponding to the surfaces of the slump model outlined in Table 5. In each plot, the quantities of all components are scaled to a 60 kg concrete batch, with each side representing a specific component and progressing in a clockwise direction around the plot. Within each plot, three components are subject to variation, while the remaining two are held constant. The optimal concrete mix is located in the upper corner of both Figure 3a, suggesting that adjusting the original component limits to increase the binder content and decrease the aggregate fractions of 0–4 mm and 4-8 mm would likely enhance the concrete slump. This is expected as the reduction in the smaller-sized fraction would result in less aggregate surface area to cover and, therefore, an increase in the amount of paste free to facilitate the flow of the mix.

A different trend emerges in Figure 3b, where the water and binder content are held constant. In this configuration, augmenting coarse aggregates’ 8–16 mm fraction content exerts the most significant influence on slump. This behavior mirrors observations in conventional concrete, where augmenting the coarse aggregate fraction up to a certain threshold enhances flowability through the previously described mechanism.

It was determined that a slump of 140 mm, as attained from the optimal concrete mix design, would be insufficient for practical applications. Consequently, as previously mentioned, the mix design was adjusted by increasing the content of the chemical activator solution until the concrete achieved a slump value of 200 mm (refer to Table 6 for details on the upscaled mix). Additionally, the substitution of limestone with olivine aggregates did not result in a reduction in slump.

### 3.5. Compressive Strength

Figure 4 illustrates the typical conical failure mode observed in the L-FNS-AAC concrete cubes. All samples exhibited elevated mean 28-day unheated compressive strengths, ranging from 65.7 MPa to 87.9 MPa, in the absence of heat curing. On average, the residual compressive strength following heating at 600 °C for 2 h amounted to approximately 35% of the unheated strength, with a range from 19% to 46%. The modification made to the DOE optimal mix to enhance slump (upscaled mix) resulted in reductions in both unheated and heated compressive strengths by 2.7% and 30%, respectively. The decline in residual compressive strength, likely attributable to the increase in water content (increasing from 164.1 kg per m^3^ to 169.6 kg per m^3^), underscores the high sensitivity of the L-FNS-AAC concrete mixes to minor variations in water content.

Figure 5 presents contour maps depicting the unheated compressive strength. Figure 5a illustrates that an increase in binder content leads to increased compressive strength when water and aggregate fraction’s 8–16 mm are held constant. The pronounced red regions on both extremes—left and right—of Figure 5b indicate that the compressive strength of L-FNS-AAC benefits (with values ranging from 85 to 90 MPa) from aggregate contents near the lower or upper limit of the 8–16 mm fraction. In contrast, intermediate values yield diminished strength, falling below 80 MPa.

The provided ternary diagrams in Figure 6 illustrate the relationship between binder content, water, and aggregates and the resulting residual compressive strength. In the first diagram (Figure 6a), increasing the binder content significantly impacts compressive strength. As the binder content increases, the compressive strength also rises. Unlike the case of unheated compressive strength, the contour map in Figure 6a shows that the relationship is not linear for the heated case. An increase in binder content outside of the optimal range, delimited by the contour at 36 (referring to 36 MPa of compressive strength), decreases this parameter. This highlights the importance of multi-component variation in AAC design, where there is a clear optimal zone.

This suggests that an optimal balance could be achieved in alternative AAC formulations prepared with different precursors, activators, and aggregates. Following a similar procedure as presented in this study would allow for the identification of optimal regions with enhanced residual compressive strength while maintaining a minimal amount of activator solution.

Figure 6b further explores the effect of water content on residual compressive strength. A similar trend can be observed: increasing the water content leads to an initial rise in compressive strength, but this effect may plateau or decrease beyond a certain point, as excessive water can dilute the binder phase and increase porosity, leading to a decrease in residual strength.

The observed decline in concrete strength can likely be attributed to the thermal incompatibility between the aggregates, in this case limestone, and the matrix. Alkali-activated matrices have a tendency to shrink at elevated temperatures, whereas aggregates tend to expand. This differential behavior results in internal stresses that, when exceeding the tensile capacity of the matrix, give rise to internal cracking, thereby compromising the concrete strength [32,36,37]. Substituting limestone with olivine aggregates mitigated the thermal mismatch between the matrix and aggregates, leading to a partial recovery (an increment of 36%) in heated compressive strength, which increased from 23.5 MPa for the upscaled L-FNS-AAC to 32 MPa for the olivine-based ferronickel slag alkali-activated concrete, O-FNS-AAC. However, this improvement in heated compressive strength was accompanied by a reduction (23%) in unheated compressive strength, decreasing from 85 MPa to 65 MPa due to the change in coarse aggregate type.

The authors determined that the properties of O-FNS-AAC are more suitable for structural engineering applications, where a 65 MPa compressive strength after 28 days is generally adequate for most applications, and the heated value of 32 MPa would often suffice (and would be preferable to the 23 MPa achieved using limestone aggregates) to support loads until post-fire assessments and repair works can be carried out.

The O-FNS-AAC residual compressive strength (50%) is comparable to what has been reported by other studies on alkali-activated concrete. For instance, Derinpinar et al. [38] observed a residual compressive strength (after exposure to 600 °C for 1 h) that was approximately 50% of the original value (35 MPa) when activating blast furnace slag using sodium hydroxide. Although the percentage retention is similar to the value reports in the present study, the absolute residual compressive strength (~17 MPa) was considerably lower than that observed in the present study (~32 MPa).

Guerrieri et al. [39] examined the residual compressive strength of three concrete mixtures: OPC-only, AAC based on slag, and a 50/50 blend of slag and OPC. All mixtures exhibited an unheated compressive strength of approximately 50 MPa after ambient curing for 28 days. Following exposure to 600 °C for 1 h, the mixtures containing slag retained around 50% of their original strength, whereas the OPC-only concrete experienced a reduction of approximately 80% in its original strength. The enhanced retention of strength in the proposed L-FNS-AAC is attributed to the absence of calcium hydroxide, which undergoes degradation at elevated temperatures and leads to diminished chemical stability in OPC-based concrete mixtures [29].

Ramagiri et al. [40] reported a strength retention of approximately 50% in AAC blends composed of slag and fly ash after subjecting the samples to a temperature of 538 °C for 2 h. The author’s best-performing blend, consisting of equal parts of slag and fly ash, exhibited compressive strengths of around 70 MPa and 35 MPa before and after heat exposure, respectively. These values align closely with those observed for the O-FNS-AAC mix evaluated in the present study. The decline in compressive strength is attributed to the degradation of the AAC matrix and the development of pore pressure, a damage mechanism corroborated by additional research [41] on AAC subjected to high temperatures, which likely contributes to the observed reduction in strength for the O-FNS-AAC mix developed in this study.

### 3.6. Life Cycle Analysis

The global warming potential (GWP) of the optimal L-FNS-AAC mix was calculated to be 113.7 kg CO_2__eq., while the upscaled mix resulted in 115.4 kg CO_2__eq. The lowest emissions were observed for mix 12, which had a poor overall performance at 94 kg CO_2__eq. per 1 m^3^ (Figure 7). In these calculations, only emissions from the ferronickel slag grinding process were considered, contributing approximately 14% to the total GWP of the mixes. The majority of emissions, 59% to 64%, were attributed to the activators (KOH and KS). The concentrations of chemical activators were varied within narrow limits, resulting in most mix designs producing similar levels of CO_2_-equivalent emissions.

When allocating emissions based on the economic value of FNS, the GWP for the optimal and upscaled mixes increased to 310 kg CO_2__eq./m^3^ and 324 kg CO_2__eq./m^3^, respectively. The primary distinction in the global emissions calculations arises from whether ferronickel slag is classified as a by-product or waste material. If classified as a by-product, emissions would be allocated based on their economic value. This would result in a higher share of emissions being attributed to FNS, as a portion of the emissions from ferronickel production would be transferred to the slag. Nevertheless, this remains a favorable scenario, as regardless of classification, there is a significant net reduction in CO_2_ emissions. Furthermore, reallocating emissions to slag could be a beneficial approach to reducing the environmental impact of the ferronickel production industry. This could also result in metallurgical industries promoting the development and use of alkali-activated concrete to benefit from reduced global emissions.

Figure 8 visually compares the CO_2_ emissions between the upscaled ferronickel slag-based alkali-activated concrete (AAC) mix design proposed in this study and a similar strength concrete based on ordinary Portland cement (OPC) produced by Chan et al. Regardless of the classification of FNS, the figure demonstrates that the AAC mix emits substantially lower CO_2_ compared to the high-strength OPC concrete developed by Chan et al. [42]. Chan’s high-strength OPC mix, which had a high OPC content, recorded a global warming potential (GWP) of 502 kg CO_2__eq. The upscaled L-FNS-AAC mix achieved a 35% reduction in GWP When FNS is classified as a by-product and a reduction of 77% GWP when classified as a waste material.

The significant reduction in CO_2_ emissions compared to equivalent OPC-based products results from the utilization of an alkaline activation combined with the optimization procedure performed based on the mixture design of experiments technique. Even when accounting for the substantial CO_2_ emissions contributed by the chemical activators, the FNS-AAC demonstrated a significantly lower environmental impact. The high CO_2_ emissions associated with OPC production arise primarily from the high energy consumption required for clinker production, which accounts for approximately 45% of the total CO_2_ emissions related to OPC cement production [43]. Kajaste et al. [44] estimated the emissions from electricity and fossil fuel consumption to be between 304 to 490 kg CO_2_ per ton of cement. The most significant portion of CO_2_ emissions is attributed to the chemical decomposition of limestone at high temperatures, generating approximately 520 kg CO_2_ per ton of cement [43].

The primary advantage of alkali activation over OPC production is the ability to produce cement without the need for burning clinker and without generating the chemical CO_2_ emissions associated with limestone calcination. However, this benefit can be compromised when large amounts of chemical activators are used. For instance, Turner et al. [45] demonstrated that using a high concentration of NaOH (16M) to produce Grade 40 concrete resulted in global warming emissions of 320 kg CO_2__eq/m^3^, which is almost identical to the emissions from OPC concrete of similar strength, recorded at 354 kg CO_2__eq.

Mix design optimization through the design of experiments is an invaluable tool for reducing the activator content and, consequently, the CO_2_ emissions associated with AAC production, as demonstrated by various studies on binder production [11,23].

## 4. Conclusions

This study presented an optimization methodology for the formulation of AAC mix design based on ferronickel slag. The binder, water, and aggregates were varied in 20 combinations to develop predictive models for the design of optimal AAC. The experimental work resulted in the following conclusions:This study demonstrates that the mixture design of experiment is an effective methodology for optimizing alkali-activated concrete (AAC) mix designs, achieving highly accurate predictive models for compressive strength and slump (less than 3.1% prediction error).A mix design for ferronickel slag-based AAC was developed with significantly reduced concentrations of chemical activators: potassium hydroxide (1.91 mol/L) and potassium silicate (1.31 mol/L), approximately four times lower than typically reported. Despite this reduction, the optimized mix achieved a compressive strength of 85 MPa and a slump of 210 mm without heat curing or additives.The best-performing L-FNS-AAC mix achieved a 28-day compressive strength of approximately 88 MPa without heat curing, indicating its suitability for high-strength concrete structures, such as bridges and high-rise buildings.The optimal mix maintained adequate workability (slump = 210 mm) for up to 3 h post-mixing. Its unheated compressive strength was comparable to the lab-scaled optimal mix, while its heated compressive strength (after 2 h at 600 °C) decreased from approximately 34 MPa to 23.5 MPa.Replacing limestone aggregates with olivine aggregates resulted in a lower unheated compressive strength (65 MPa, 23% lower than limestone) but improved residual compressive strength (32 MPa, 36% higher than limestone) while maintaining a slump of 200 mm.A life cycle analysis assessed the global warming potential of all L-FNS-AAC mixes. Assuming zero economic allocation for ferronickel slag, the alkali activator chemicals (KOH and KS) contributed approximately 60% to GWP. Compared to an equivalent OPC-based concrete, the upscaled L-FNS-AAC mix resulted in 35% to 77% lower CO_2_-eq. emissions, depending on whether ferronickel slag is considered an industrial by-product or waste material.

The use of mixture design of experiments, as presented in this study, provides an effective tool to address the challenge of lack of standardization in the field of AAC mix design. Additional challenges associated with adopting alkali-activated concrete (AAC) include the alkali-silica reaction, which can be significantly reduced by incorporating calcium nitrate [46]. The presented results emphasize the importance of multi-component optimization in AAC design to achieve desired mechanical properties while minimizing the environmental impact of chemical activators. This research presents a promising pathway for developing efficient, sustainable and durable AAC solutions in construction where the current lack of standards and durability data has slowed the adoption of AAM technology

## Figures and Tables

**Figure 1 materials-17-04939-f001:**
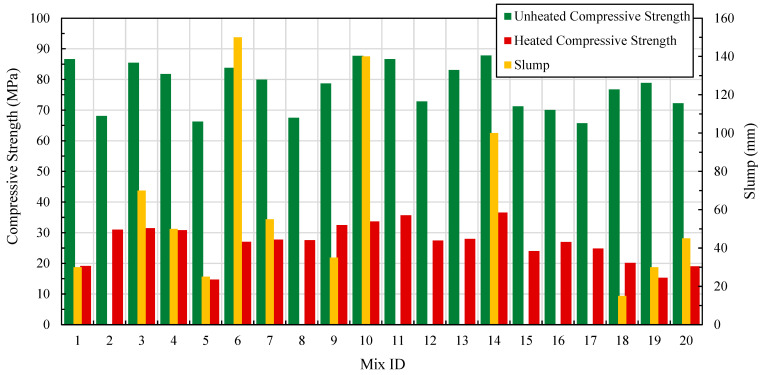
Slump and compressive strength results for all L-FNS-AAC (DOE concrete mixes).

**Figure 2 materials-17-04939-f002:**
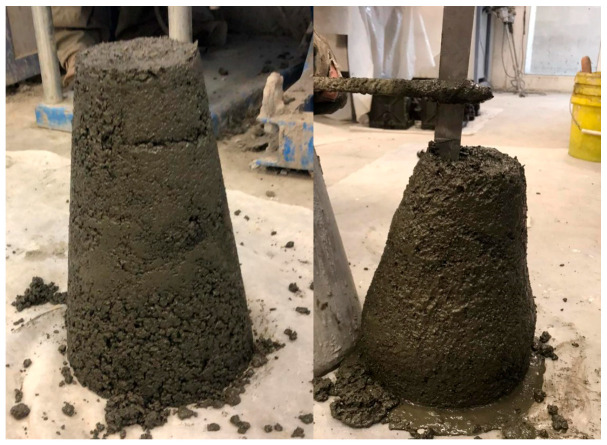
Slump test of fresh concrete for mixes 8 and 3.

**Figure 3 materials-17-04939-f003:**
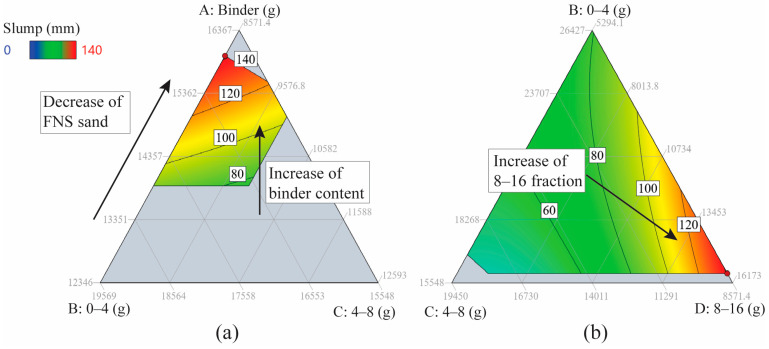
2D ternary contour plots for slump as a function of three components: (**a**) A, B, C with fixed water and coarse aggregates 8–16 mm; (**b**) A, C, D with water and FNS sand content fixed.

**Figure 4 materials-17-04939-f004:**
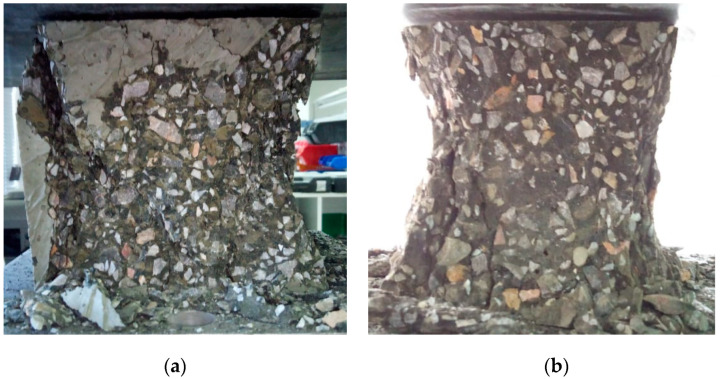
Post failure cube samples of L-FNS-AAC: (**a**) unheated; (**b**) heated samples.

**Figure 5 materials-17-04939-f005:**
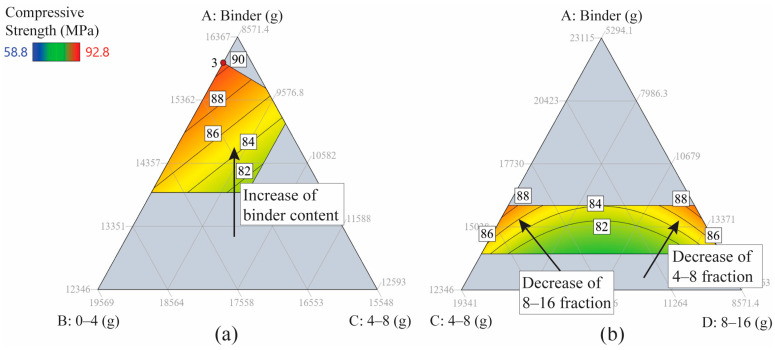
2D ternary contour plots for unheated compressive strength as a function of three components: (**a**) A, B, C with fixed water and coarse aggregates 8–16 mm; (**b**) A, C, D with water and FNS sand content fixed.

**Figure 6 materials-17-04939-f006:**
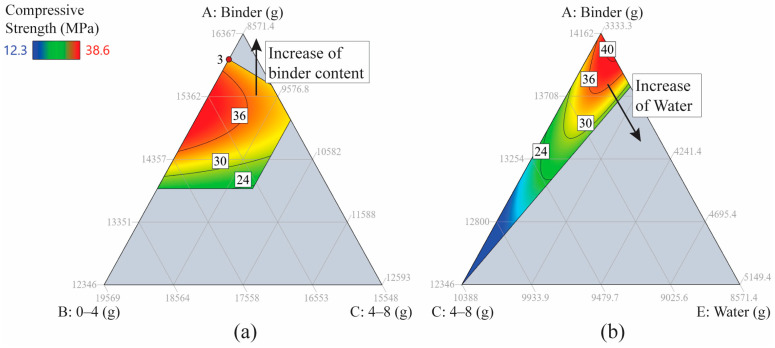
2D ternary contour plots for heated compressive strength as a function of three components: (**a**) A, B, C with fixed water and coarse aggregates 8–16 mm; (**b**) A, C, E with FNS sand and coarse aggregate 8–16 mm content fixed.

**Figure 7 materials-17-04939-f007:**
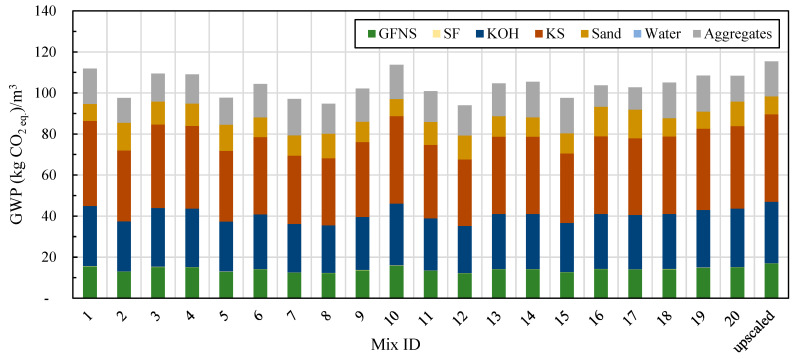
Global Warming Potential of L-FNS-AAC mixes (upscaled mix included), where no allocation is applied to FNS.

**Figure 8 materials-17-04939-f008:**
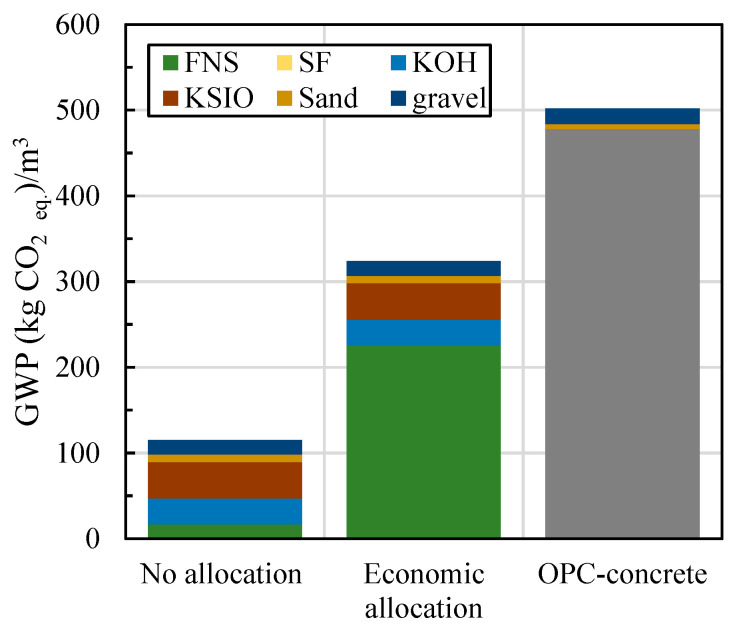
Impact of allocation method on total GWP of L-FNS-AAC (upscaled mix) compared to a strength-wise equivalent OPC-based concrete.

**Table 1 materials-17-04939-t001:** Ferronickel slag and silica fume chemical analysis through XRF by weight *.

Precursor	SiO_2_ (%)	Al_2_O_3_ (%)	CaO (%)	Fe_2_O_3_ (%)	MgO (%)	Na_2_O (%)	P_2_O_5_ (%)	K_2_O (%)	TiO_2_ (%)	MnO (%)	LOI-Flux
GFNS	36.9	3.61	4.18	32.8	7.41	0.15	0.02	0.48	0.19	0.00	0.00
SF	88.9	0.73	0.34	1.01	0.63	0.71	0.03	1.50	0.00	0.12	6.82

* Only detectable chemical compounds are listed.

**Table 2 materials-17-04939-t002:** CO_2_-eq., energy consumption, and component limits for GFNS concrete mixes with limestone aggregates.

Material	Code	CO_2-_eq_._ (kg/t)	Energy Consumption (MWh/t)	Mixture Boundaries, by Weight (%)	
				Lower Bound	Upper Bound
Binder	A	100.41 *	0.533	20.6%	26.6%
FNS sand 0–4 mm	B	12.93	0.052	25.9%	45.3%
Limestone aggregates 4–8 mm	C	12.55	0.051	14.3%	30.4%
Limestone aggregates 8–16 mm	D	12.55	0.051	8.8%	28.6%
Water	E	0.37	0.002	5.6%	6.3%

* Only the emissions related to the grinding process were considered for GFNS.

**Table 3 materials-17-04939-t003:** L-FNS-AAC (DOE concrete mixes) composition, per m^3^.

Mix ID	Water (kg)	GFNS (kg)	SF (kg)	KOH (kg)	KS (kg)	FNS Sand 0–4 (kg)	Limestone Aggregates 4–8 (kg)	Limestone Aggregates 8–16 (kg)	Water * to Binder **	KOH (mol/L)	KS [mol/L]	SiO_2_/K_2_O
1	159.9	575.8	42.5	18.5	43.8	682.7	759.7	351.7	0.235	2.06	1.40	1.26
2	152.4	479.9	35.4	15.4	36.5	1119.4	548.8	231.1	0.269	1.80	1.22	1.29
3	156.8	564.5	41.6	18.1	43.0	919.9	449.1	429.7	0.235	2.06	1.40	1.26
4	164.9	559.4	41.3	18.0	42.6	896.7	564.0	351.0	0.249	1.94	1.32	1.27
5	144.7	478.4	35.3	15.4	36.4	1050.4	372.0	471.5	0.256	1.89	1.28	1.28
6	164.1	523.3	38.6	16.8	39.8	800.7	775.3	267.2	0.265	1.83	1.24	1.28
7	147.4	462.0	34.1	14.9	35.2	823.0	379.1	758.3	0.270	1.80	1.22	1.29
8	144.9	454.1	33.5	14.6	34.6	987.3	529.7	409.8	0.270	1.80	1.22	1.29
9	161.8	507.1	37.4	16.3	38.6	816.7	505.6	535.1	0.270	1.80	1.22	1.29
10	164.1	590.7	43.6	19.0	45.0	698.1	375.0	689.6	0.235	2.06	1.40	1.26
11	145.9	497.5	36.7	16.0	37.9	924.3	375.1	592.1	0.248	1.95	1.32	1.27
12	143.7	450.3	33.2	14.5	34.3	961.6	706.8	242.2	0.270	1.80	1.22	1.29
13	145.6	524.2	38.7	16.9	39.9	827.8	796.5	231.2	0.235	2.06	1.40	1.26
14	156.6	524.2	38.7	16.9	39.9	773.6	380.5	733.3	0.253	1.92	1.30	1.27
15	145.5	470.5	34.7	15.1	35.8	808.2	754.9	354.9	0.262	1.85	1.26	1.28
16	146.1	525.9	38.8	16.9	40.0	1190.8	438.6	232.0	0.235	2.06	1.40	1.26
17	164.3	519.4	38.3	16.7	39.5	1160.9	375.5	314.1	0.268	1.81	1.23	1.29
18	145.8	524.8	38.7	16.9	39.9	744.8	477.4	635.6	0.235	2.06	1.40	1.26
19	157.2	550.8	40.6	17.7	41.9	693.7	656.4	470.0	0.241	2.01	1.36	1.26
20	155.1	558.5	41.2	18.0	42.5	986.3	580.0	230.5	0.235	2.06	1.40	1.26

* Water stands for the sum of the water from silicate solutions and additional water used. ** Binder stands for the sum of GFNS, SF, KOH, and the dry part of KS.

**Table 4 materials-17-04939-t004:** Actual model-coded coefficients for slump and compressive strength before and after heat exposure (all ×10^−5^) *.

Parameter	A	B	C	D	E	AB	AC	AD	AE	BC	BD	BE	CD	CE	DE	ABC	ABE	ACD
Slump [mm]	−39,124	−5863	−3023	−6598	−3,170,217	0.2277			66.50			57.09	−0.1065	57.36	58.66			
Compressive Strength ^1^ Unheated [MPa]	342.2	−204.0	361.8	95.85	602.30						0.02031		−0.0380					
Compressive Strength Heated [MPa]	−142,200	−54,800	−2113	18,872	−1,532,843	6.632	2.618	1.348	54.99	1.161	0.03984	39.29	0.6475	21.21	20.45	−0.00007879	−0.001351	−0.00004490

* Coefficients fields left empty have a value of zero. ^1^ The compressive strength was measured after 28 days.

**Table 5 materials-17-04939-t005:** Fit statistics for slump and compressive strength models.

Response	Mean	Std. Dev.	CoV %	R^2^	Adjusted R^2^	Predicted R^2^	Adequate Precision
Slump (mm)	33.1	6.08	18.4	0.99	0.98	0.90	30.3
Compressive strength ^1^ unheated (MPa)	77.8	4.90	6.3	0.69	0.65	0.59	15.3
Compressive strength heated (MPa)	26.6	3.62	13.6	0.83	0.75	0.63	10.5

^1^ The compressive strength was measured after 28 days.

**Table 6 materials-17-04939-t006:** Optimized (predicted), measured and modified (upscaled) L-FNS-AAC mix designs and errors in predictions.

Optimal Concrete Mix	Water (kg)	GFNS (kg)	SF (kg)	KOH (kg)	KS (kg)	Aggregate (FNS Sand, Limestone 4–16 mm)			Slump (mm)	Compressive Strength ^1^	
						0–4 (kg)	4–8 (kg)	8–16 (kg)		Unheated (MPa)	Heated (MPa)
Predicted	164.1	590.7	43.6	19	45	698.1	375	689.6	141.4	90.5	33.6
Measured ^2^	164.1	590.7	43.6	19	45	698.1	375	689.6	140	87.7	33.7
Difference (%)	0	0	0	0	0	0	0	0	−1.00	−3.10	0.30
Upscaled	169.6	604.6	44	18.2	43.4	692.5	373.7	685.2	210	85.3	23.5
Difference (%)	3.37	2.30	0.80	−4.00	−3.50	−0.80	−0.30	−0.60	50.0	1.50	−30.3

^1^ The compressive strength was measured after 28 days. ^2^ The measurement corresponds to mix 10 which had an identical composition to the optimal formulation predicted by DOE.

## Data Availability

The original contributions presented in the study are included in the article. Further inquiries can be directed to the corresponding author/s.

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
