# Peer review of "Performance-Based Design of Ferronickel Slag Alkali-Activated Concrete for High Thermal Load Applications"

_materials, 2024, doi:10.3390/ma17194939_

Round 1

Reviewer 1 Report

Comments and Suggestions for Authors

Title: Performance Based Design of Alkali Activated Concrete for High Thermal Load Applications.

Decision: Major revisions.

General comments:

1. The title highlights the use of ferronickel slag in the research. This is clear just by reading the article, but it is an important point that needs to be highlighted from the title. In the current format, the title is generic, please review this.

2. The abstract of the manuscript is very poor, it does not present the main objectives, main results and conclusions. Please revise this section appropriately.

3. Introduction: the authors take the classic approach for this type of work, highlighting that the production of OPC is very harmful to the environment, and highlighting the use of AAM as an alternative. However, note that the production of AAM uses activators such as silicate and hydroxides, and is also a material with CO2 emissions. Please highlight this in the text of the manuscript. Furthermore, it is important to be critical in the manuscript and include the main limitations or disadvantages of using AAM, reported in the bibliography, such as: low workability and problems in rheological behavior when compared to OPC, highlighted for example in Silica fume activated by NaOH and KOH in cement mortars: Rheological and mechanical study; problems related to the need for thermal curing and high alkalinity of the activating solution when using slags as a precursor, which can increase the cost of the product and make it difficult to use labor; and related problems, as highlighted in Activated alkali cement based on blast furnace slag: effect of curing type and concentration of Na20. Please discuss this information in the text of the manuscript. The authors are invited to make clear in the text the difficulties of using AAM, which still make the application of this material on a real scale as a substitute for OPC unfeasible.

4. The authors highlight several works in the introduction, such as Kharazi et al. [8], Simsek et al. [9], Dai et al. [10] and others. This approach is interesting because the authors highlight information collected in other studies. However, I feel that the authors should not make clear in the text of the manuscript what the differences are between these studies and the results evaluated by the authors. Please make clear in the text of the manuscript what the main novelties of the research are and how its results are original contributions.

5. The authors are evaluating the effect of ferronickel slag. However, I would like the authors to include more specific information about this, such as other works with similar research, data on the generation of this type of slag, chemical and mineralogical differences in reaction to blast furnace slag widely used in this type of material. Please review this information and change it appropriately according to the recommendations.

6. “Ground ferronickel slag served as the primary aluminosilicate source material.” I do not agree with this statement. Note that the SiO2 content = 36.90% and Al2O3 = 3.61%. However, the Fe2O3 content = 32.8%, higher than the aluminate content. How is this material a source of aluminosilicate? This is wrong. It is known that some alkali-activated materials and/or geopolymers are based on ferrosialates, which seems to be your case. I think the dosing information presented by the authors is completely incorrect if the material is considered as a source of aluminosilicate. Please explain this information appropriately.

7. Is Figure 1 necessary? Please explain.

8. The authors do not make clear the dosing criteria for the AAM used in the research. Please specify parameters of the activating solution, such as molarity, silica modulus, molar concentrations; if the work is about AAM, this information needs to be well worked on in the text. Use bibliographical references to justify the dosing performed in the research. Otherwise, the article is not scientific.

9. Is Table 1 necessary? Why not present the results obtained based on the mean and deviations in the form of figures? This way, the effect of the material on the properties of AAM is clearer. Please consider this.

10. A critical point of the research is that it is not possible to understand the pattern of results obtained in the research. For example, in Table 1 we observe the difference in slump and compressive strength results for the dosages performed in the research. Although it is possible to observe the discrepancy in results, it is not clear which parameters affect these discrepancies. For example, MIX 1 presents an Unheated compressive strength of 85.5 MPa, while MIX 2 presents 68.1 MPa for the same property. What justifies this discrepancy in results? Is it the parameters of the activator solution? Such as molarity or silica modulus? Is it related to the aggregate or packing content? Is it due to the GFNS content used? This is not clear, there is no such discussion in the article. In other words, although there are many experimental results, the authors do not take the care to explain to the readers of the manuscript which parameters justify the discrepancy in the information obtained. Please review the text of the manuscript and change this information appropriately.

11. “Many of the DOE mixes yielded zero slump, a condition likely attributable to the stringent limits set for water content (ranging from 5.6% to 6.3% by weight). Additionally, this outcome can be associated with the elevated silica modulus (1.6), which Law et al. [22] identified as a contributing factor to slump loss.” Although the authors justify this information by the silica modulus, note that at no point in the article is it easy to examine this information for the mixes defined in their research. This needs to be changed in the text of the manuscript. Please revise this accordingly.

12. Figure 3 is very confusing and difficult to review. It is not clear what is the text discussing the results and what is the figure legend. In addition, there seem to be several Figure 3s in the text, which makes it difficult to understand the discussion proposed by the authors. In addition, there are Figure 3s that are cropped without any explanation, the image quality is low, and the text is difficult to read. The authors need to completely revise this part: standardize the figures presented without cropping, with visible scales and letters, with appropriate captions, without repeated numbers, and with a text discussing the results. Otherwise, it is difficult to understand the discussion proposed by the authors. Please revise appropriately.

13. 3.6 Life Cycle Analysis: This is an important part of your article, however I feel that the authors do not discuss the results adequately. There is no comparison of results in the research. This methodology has already been used by other authors, including in the analysis of AAM. Therefore, it is necessary for the authors to revise this part of the text and discuss the information appropriately, taking into account other similar works and research. Compare the information and results obtained in your article.

14. The conclusions are poor. First revise the information about the mixes used in the research and then appropriately change the conclusions taking these changes into account. This way, it will be clearer to future readers of the article the effects of the main AAM parameters on the results obtained in your research.

Reviewer 2 Report

Comments and Suggestions for Authors

This is a very interesting paper about the high temperature resistance of alkali-activated concrete.  A response surface methodology is adopted considering several concrete properties. Generally speaking, the experiments are well designed and the conclusions are well supported. Below are my comments.

1. For design of alkali-activated materials, it is worth clarifying that there are generally two methods: (1) preparing alkaline solution with specific concentration and considering solid-to-liquid ratio; (2) considering water-to-binder ratio where the binder comprises both precursors and Na2O. The second method will control the water content in different mixtures while the first method is easier to be used.

2. The introduction should include the limitations of literatures and the specific innovations of this study.

3. You are focusing on high temperature resistance but the introduction only includes mix design.

4. Why are 600 C and 2 hours selected?

5. Have you considered the energy consumption in the grinding process in LCA?

6. In the conclusion part, please take care of "This section is not mandatory but can be added to the manuscript if the discussion is unusually long or complex."

7. The potential limitation should be discussed. The durability properties are not discussed. Recent papers have pointed out alkali-activated concrete may still have alkali-silica reaction. The ASR could be more serious when pH is high which favors the silicon dissolution. Recent published papers should be reviewed. ('Calcium nitrate effectively mitigates alkali-silica reaction by surface passivation of reactive aggregates').

Reviewer 3 Report

Comments and Suggestions for Authors

Authors have conducted some experiments and LCA in this research. However, there are some observations on the research.
1- Line 145 "Concrete Production & Specimens' production" Please change it to -Concrete Production and specimens preparation

2- Line 195 "To facilitate calculations the authors relied on Design Expert® v11.1.2.0. was selected." Please re-write for better understanding

3- Why 600 oC? Please justify your selection

4- The heating protocol was standardized, with the temperature increment set at 5°C/min until the desired temperature was attained. Which standard? Any reference for this thermal loading rate?

5- the heating was maintained for a duration of 2 hours...Why 2 hours, why not 3, 4 hours? Please provide some references on your selection or proper reasoning

6- Table 4, in some cases, slump value was 0 and in some cases it was 150mm. Do you think compaction of 30 seconds was enough for both type of concrete with 0 slump and 150 mm slump? Please justify your answer with some results for example density or porosity test... It can also be seen that the specimen with lower slump has lesser Compressive strenth (Table 4) as compared to the one with higher slump...This indicates that specimens with lower slump value has higher porosity. Therefore, compaction of just 30s for all type of specimen is not justified. This research lack some critical points to be discussed before final acceptance

7- Table 4, please provide the graphical presentation also for reader's quick -knowledge and understading

8- Figure 3, Please provide picture with good resolution. It's difficult to read on the figure

9- What is the justification to use cube samples but not cylindrical specimens for these test?
10- Results are not enough to have the comprehensive documents. Author's should select one of the best mix design composition and perform other tests such as tensile test, flexure test and other durability related tests to provide some useful and in-depth knowledge. There are several studies available on heating and compression strength tests. 
11- Please highlight the novelty of this research

12- This study lack some experimental results. Without including those results, it is not comprehensive and hence difficult to give the positive opinion on it.

Round 2

Reviewer 1 Report

Comments and Suggestions for Authors

Accept in present form

Author Response

Thank you very much for your positive feedback and for recommending the acceptance of our manuscript. We appreciate your time and effort in reviewing our work.

Reviewer 2 Report

Comments and Suggestions for Authors

This paper has been revised and should be good for publication.

Author Response

Thank you for your supportive feedback. We are pleased to hear that you find the revised manuscript suitable for publication. We appreciate your efforts in reviewing our work.

Reviewer 3 Report

Comments and Suggestions for Authors

Previous Comment: Line 145 "Concrete Production & Specimens' production" Please change it to -Concrete Production and specimens preparation... I cannot see the modification. Please correct it.

Author Response

Comment:

Previous Comment: Line 145 "Concrete Production & Specimens' production" Please change it to -Concrete Production and specimens preparation... I cannot see the modification. Please correct it.

Reply:

Thank you for your careful review. We apologize for the oversight. The suggested change from "Concrete Production & Specimens' production" to "Concrete Production and specimens preparation" has now been implemented in line 220 of the updated manuscript, heading 2.2.